# Genome-Wide Analysis of the DNA Methylation Profile Identifies the fragile histidine triad (*FHIT*) Gene as a New Promising Biomarker of Crohn’s Disease

**DOI:** 10.3390/jcm9051338

**Published:** 2020-05-04

**Authors:** Tae-Oh Kim, Dong-Il Park, Yu Kyeong Han, Keunsoo Kang, Sae-Gwang Park, Hae Ryoun Park, Joo Mi Yi

**Affiliations:** 1Department of Internal Medicine, Inje University, Haeundae Paik Hospital, Busan 48108, Korea; kto0440@paik.ac.kr; 2Department of Internal Medicine, Kangbuk Samsung Hospital, Sungkyunkwan University School of Medicine, Seoul 03181, Korea; diksmc.park@samsung.com; 3Department of Microbiology and Immunology, College of Medicine, Inje University, Busan 47392, Korea; parkha74@inje.ac.kr (Y.K.H.); micpsg@gmail.com (S.-G.P.); 4Department of Microbiology, Dankook University, Cheonan 31116, Korea; kangk1204@gmail.com; 5Department of Oral Pathology, School of Dentistry, Pusan National University, Yangsan, Gyeongsangnam do 50612, Korea; parkhr@pusan.ac.kr

**Keywords:** DNA methylation profile, Crohn’s disease (CD), hypermethylation, gene network

## Abstract

**** Inflammatory bowel disease is known to be associated with a genetic predisposition involving multiple genes; however, there is growing evidence that abnormal interactions with environmental factors, particularly epigenetic factors, can also significantly contribute to the development of inflammatory bowel disease (IBD). Although many genome-wide association studies have been performed to identify the genetic changes underlying the pathogenesis of Crohn’s disease, the role of epigenetic alterations based on molecular complications arising from Crohn’s disease (CD) is poorly understood. We employed an unbiased approach to define DNA methylation alterations in colonoscopy samples from patients with CD using the HumanMethylation450K BeadChip platform. Technical and functional validation was performed by methylation-specific PCR (MSP) and bisulfite sequencing of a validation set of 207 patients with CD samples. Immunohistochemistry (IHC) analysis was performed in the representative sample sets. DNA methylation profile in CD revealed that 135 probes (24 hypermethylated and 111 hypomethylated probes) were differentially methylated. We validated the methylation levels of 19 genes that showed hypermethylation in patients with CD compared with normal controls. We uniquely identified that the fragile histidine triad (*FHIT*) gene was hypermethylated in a disease-specific manner and its protein level was downregulated in patients with CD. Pathway analysis of the hypermethylated candidates further suggested putative molecular interactions relevant to IBD pathology. Our data provide information on the biological and clinical implications of DNA hypermethylated genes in CD, identifying *FHIT* methylation as a promising new biomarker for CD. Further study of the role of *FHIT* in IBD pathogenesis may lead to the development of new therapeutic targets.

## 1. Introduction

Inflammatory bowel diseases (IBDs), containing ulcerative colitis (UC) and Crohn’s disease (CD), are significantly heterogeneous diseases. Despite significant progress in understanding the molecular mechanisms involved in the development of IBD, the exact pathogenesis remains unclear [1]. Recent literature supports the theory that IBD is the result of an abnormal immune response against commensal bacteria and luminal antigens in a genetically susceptible host [2]. Genetic studies, including linkage mapping analysis and genome-wide association studies (GWAS), have improved our understanding of the importance of genetic susceptibility in IBD, and more than 30 risk-associated loci have been identified by meta-analyses [3,4]. However, these genetic risk factors can only explain approximately 20% of the disease risk [5], suggesting that epigenetic factors are possibly involved in the pathogenesis of IBD [6].

Transcriptional gene silencing by promoter DNA methylation is a critical epigenetic alteration in human cancers, as well as other diseases. Particularly in cancer, abnormal DNA methylation occurs in promoter CpG islands, resulting in the inactivation of tumor suppressor genes [7]. The involvement of DNA methylation in many different cancers and other human diseases, including IBD, has been broadly studied [8]. Inflammation is a common component of many of these diseases. An early study showed that high-grade dysplasia in inflamed colonic mucosa was associated with the hypermethylation of specific genes, supporting the hypothesis that chronic inflammation is associated with increased levels of DNA methylation [9]. In UC, patient aberrant DNA methylation has been demonstrated in the estrogen receptor, *p14*ARF, and *E-cadherin* genes, and growing evidence suggests the many genes regulated by epigenetic mechanisms are involved in the pathogenesis of IBD [10,11]. Additionally, we previously screened the CpG islands in the promoter region of several genes that are known to be highly methylated in colon tumor tissues and in tissues specifically obtained from patients with IBD [12,13]. Very recently, several genes showed significant evidence of differential methylation patterns in IBD and many more of these genes than expected by chance overlapped with genes that were previously implicated in IBD susceptibility in GWAS [14]. However, little is known regarding genome-wide DNA methylation changes in patients with CD. DNA methylation analysis is leading to a brand new generation of cancer biomarkers [15]. Very recently, abnormally methylated DNA sequences were detected in circulating DNA in the blood of cancer patients by conventional methylation-specific PCR (MSP) and had high sensitivity and specificity compared with tumor tissues [16,17].

To understand the mechanism of the pathogenesis of IBD, we aimed to evaluate the impact of differential methylation patterns in tissue biopsies from patients with CD. For this study, we used the Illumina HumanMethylation450K Beadchip array platform, which covers 99% of RefSeq genes [18], and this technology facilitates extensive analysis of differential DNA methylation. We applied this technology to identify differential DNA methylation changes in patients with CD. Significantly, DNA hypermethylated candidate genes in CD were connected with meaningful cellular pathways using a gene-network database. Our data provides new potential methylation biomarkers for CD and suggests that detecting DNA methylation in patients with CD can be useful in clinical assessment for the diagnosis or prognosis of CD.

## 2. Experimental Section

### 2.1. Patients and Clinical Samples

The CD and normal colon biospecimens for this study were provided by the Inje Biobank (nje University, School of Medicine), a member of the National Biobank of Korea, which is supported by the Ministry of Health and Welfare. Tissue samples were collected from the active inflammatory site at the time of colonoscopy. All patients (*n* = 12) had a confirmed diagnosis of CD based on their clinical symptoms and endoscopic and pathological records and healthy individuals (*n* = 12) with no evidence of gastrointestinal disease or family history of IBD provided by the Inje Biobank were used as controls. In addition, we used a large number of patients with CD (*n* = 207) to validate the methylation of the fragile histidine triad (*FHIT*) gene. The main clinicopathological characteristics of the patients with CD for experimental validation are described in Table 1. This study was approved by the respective institutional review boards of the participating institutions of the National Biobank of Korea (IRB No: 129792-2015-056) and Kangbuk Samsung Hospital (IRB No 2016-07-029), and written informed consent was obtained from all study participants prior to data collection.

### 2.2. HumanMethylation450K BeadChip Array Analysis

The genome-wide DNA methylation levels of approximately 480,000 CpG sites were determined using the Infinium HumanMethylation450K BeadChip kit (Illumina), according to the manufacturer’s instructions. Raw data were processed using the GenomeStudio application by Macrogen (Macrogen Inc., Seoul, Korea). The normalized MethyLumiSet function implemented in the methylumi package [19] using R was used to correct background signal and dye bias. Differentially methylated CpG sites were then identified using the lmFit, eBayes, and topTable functions implemented in the limma package [20] with an adjusted (Benjamini-Hochberg) *p*-value cutoff of 0.001.

### 2.3. Methylation Analyses; MSP and Semi-Quantitative MSP

DNA was extracted from all patients with CD and normal tissues we tested in this study, following a standard phenol-chloroform extraction. Bisulfite modification of genomic DNA was carried out using the EZ DNA Methylation kit (Zymo Research, Irvine, CA, USA). For MSP and quantitative methylation analyses, we performed the same procedure as previously published [21]. The primer sequences are listed in Appendix A.

### 2.4. Bisulfite Sequencing

One microgram of genomic DNA from each sample was bisulfite-converted using the EZ DNA Methylation kit (Zymo Research, Irvine, CA, USA), following the manufacturer’s protocol. Bisulfite-modified DNA was amplified and PCR amplicons were gel-purified and sub-cloned into the pCRII-TOPO vector (Invitrogen, Carlsbad, CA, USA). At least 5–7 clones were randomly selected and sequenced on an ABI3730xl DNA analyzer to ascertain the methylation patterns of each locus. The primer sequences for bisulfite sequencing are listed in Appendix A.

### 2.5. Immunohistochemistry

Formalin-fixed, paraffin-embedded specimens of patients with CD and normal colon tissues were sectioned at a 5 μm thickness. The sections were deparaffinized in xylene and rehydrated through graded alcohol solutions. Heat-induced antigen retrieval was performed in a microwave oven for 10 min in Tris/EDTA buffer (pH 9.0). Endogenous peroxidase activity was inactivated by treatment with 3% H_2_O_2_ in PBS for 15 min. After blocking nonspecific binding (0.75% BSA in PBS) the sections were incubated with anti-FHIT (1:100, LSBio, Seattle, DC, USA) overnight at 4 °C. The staining was visualized by a peroxidase-conjugated secondary antibody and diaminobenzidine (Vector labs, Burlingame, CA, USA). Finally, the sections were counterstained by Mayer’s hematoxylin and mounted and photographed with an Axioplan microscope (Carl Zeiss, Oberkochen, Germany). Primary antibodies were omitted for the negative controls.

### 2.6. Gene Ontology and Network Analysis

Gene ontology analysis of hypermethylated and hypomethylated CpG site-associated genes in CD compared to normal samples was conducted using Metascape http://metascape.org/gp/index.html/main/step1 [22]. Network analysis of hypermethylated CpG site-associated genes was performed using the GeneMania application [23] via Cytoscape [24]. Coexpression, physical interactions, colocalization, genetic interactions, shared protein domains, and predicted interactions were considered for the network analysis.

### 2.7. Statistical Analysis

All statistical analyses were performed using the STATA 9.2 software package (College Station, TX, USA). Most analyses were conducted using a *t*-test, whereas continuous variables were analyzed using the Mann–Whitney *U* test. *p* values less than 0.05 were considered significant.

## 3. Results

### 3.1. Differential DNA Methylation Profiles in Patients with CD Compared to Normal Colon Tissues

To characterize genome-wide patterns of DNA methylation in patients with CD compared with normal controls, colonoscopy samples from patients with CD and normal colon tissues were analyzed using the HumanMethylation450K Beadchip platform (Illumina), which included 454,215 CpG sites out of a total 485,577. The methylation level at the CpG site is measured by means of a continuous variable. Using this platform, we screened differential DNA methylation patterns of normal colon controls (*n* = 3) and patients with CD (*n* = 8) to determine loci with differential DNA methylation, mostly occurring in promoter regions between patients with CD and normal controls.

We determined significant increases (hypermethylation) or decreases (hypomethylation) in methylation levels in patients with CD compared to controls with an adjusted (Benjamini–Hochberg) *p*-value cutoff of 0.001. We further filtered out hypermethylated probes in patients with CD samples by using strict criteria (>2-fold change of β-values in patients with CD compared to normal colon tissues).

A total of 2016 hypermethylated probes and 1162 hypomethylated probes were identified that represented differential methylation patterns in patients with CD when compared with normal colon tissues (Figure 1a). Based on the global distribution of hypermethylation and hypomethylated loci, we analyzed the frequency of loci in promoter, intergenic, and intragenic regions (Figure 1b). Promoter regions were defined as −1500 bp to transcriptional start sites and included the exon 1 region, and intergenic regions were defined as the regions of the genome not covered by the other classes. In the methylation profiles of patients with CD, differentially methylated regions were located in multiple gene regions and ordered as follows: promoter region (35%) or gene body (35%) > intergenic region (27%) > 3′UTR (3%) for hypermethylated regions; and intergenic region (35%) > promoter region (32%) > gene body (31%) > 3′UTR (2%) for hypomethylated regions. DNA methylation within the gene body has previously been reported to have an important role in transcriptional control [25].

It is well-established that methylated promoter CpG islands (CGIs) are associated with gene transcriptional silencing [7]. Another report suggested that DNA methylation alterations outside CGIs located in shelves or shore regions could change gene transcription and may be tissue- or cell type-specific [26]. We therefore tested whether CGI methylation associated with transcriptional silencing is a feature of CD (Figure 1b). The majority of differential DNA methylation was found outside CGIs. For differentially methylated DNA regions that were hypermethylated and hypomethylated, 5% and 4% of regions were seen in CGIs, 32% and 19% of regions were in shore regions (2-kb regions flanking CpG islands), and 8% and 8% of regions were in the shelves (2-kb regions of flanking shore regions), respectively.

Using stricter cutoff criteria (more than 3-fold for hypermethylation and less than 0.3-fold for hypomethylation in CD samples, as compared with the methylation profiles of normal controls), hierarchical clustering showed 135 CpG sites (24 hypermethylated and 111 hypomethylated) that were differentially methylated (Figure 1c). The genomic distribution of these 135 differentially methylated CpG sites (a stringent criterion) was similar to that of the 3178 differentially methylated CpG sites (which were identified with less stringent criteria) in Figure 1b, except for the hypermethylated CpG sites (Appendix A). However, this might be due to the small number (*n* = 24) of hypermethylated CpG sites when the stringent criteria were used.

### 3.2. Selected Hypermethylated Candidate Genes in Patients with CD

Abnormal DNA methylation in human diseases has been the most studied epigenetic modification during the last decade [7]. Recently, aberrant DNA methylation has been implicated as a critical mechanism in the pathogenesis of IBD. Regarding differential DNA methylation changes in the tissues of patients with CD, compared with normal colon tissues, we analyzed differentially methylated genes, which include classical CpG islands in promoter regions.

We therefore questioned the mechanism of the epigenetic impact of hypermethylated genes in CD, so we focused on identifying hypermethylated genes in patients with CD. We profiled hypermethylated genes in patients with CD, compared to normal colon tissues, and revealed a differential methylation pattern based on hierarchical clustering analysis. Among hypermethylated genes in patients with CD, we selected candidate genes (*ZFP36L1, ANXA2, EP400, FHIT, TPPP, IL5RA, MDFIC, MUM1, PUSL1, RUNX3, C19orf24, TRPM4, PPP1R15A, CDT1, SFRS1, EPHA4, KBTBD11, CCDC42B,* and *HNRNPUL1*) that were hypermethylated (>3-fold increase in methylation) (Appendix A) in patients with CD samples (Figure 2a). We determined the methylation levels of these 19 genes between normal samples and patients with CD samples from the methylation profile and indeed found that most of the 19 candidate genes were significantly hypermethylated in patients with CD, compared to normal controls (Figure 2a).

We aim to validate whether these 19 genes hypermethylated in a handful of patients with CD (*n* = 12) and healthy normal samples (*n* = 12). Two patients had colitis and 10 patients had ileocolitis, and the disease behavior was inflammatory in all 12 patients. Samples from healthy individuals, with no evidence of gastrointestinal disease or family history of IBD, provided by the Inje Biobank, were used as controls. We next designed MSP primers located in the CpG islands of these 19 genes and performed MSP analysis (Appendix A). Next, a massive MSP analysis was performed to analyze the promoter methylation patterns of the candidate genes in normal controls and patients with CD (Figure 2b). First, two (*IL5RA* and *CCDC24B*) out of the 19 genes were eliminated due to the lack of CpG islands in their promoter region. Second, nine genes (*TRPM4*, *MUM1*, *EPHA4*, *MDFIC*, *TPPP*, *EP400*, *PPP1R15A*, *HNRNPUL1*, and *C19orf24*) were found to be hypermethylated in normal tissues, indicating the lack of a CD-specific methylation pattern. Third, seven genes (*SFRS1*, *CDT1*, *ZFP36L1*, *RUNX3*, *KBTBD11*, *ANXA2*, and *PUSL1)* showed no methylation in patients with CD tissues. We finally observed that only the *FHIT* gene exhibited lower methylation in normal tissues, but was frequently hypermethylated in CD samples (Figure 2b).

### 3.3. The FHIT Gene: A CD-Specific Hypermethylated Candidate Gene and its Biological Implications

We examined the CpG islands of the *FHIT* gene in the UCSC database and found that a typical CpG island was located in the promoter region of the fragile histidine triad (*FHIT*) gene upstream region (Figure 3a). For methylation analysis of the *FHIT* gene, we designed MSP primer sets in this CpG island region. Interestingly, CpG islands in the promoter region of the FHIT gene showed a lack of methylation in normal controls and hypermethylation in patients with CD, suggesting that this gene exhibited a pattern of CD-specific methylation (Figure 3b, left panel).

To determine the impact of hypermethylation of *FHIT* in patient with CD samples, we expanded our methylation analyses of the *FHIT* gene to a larger cohort of patients with CD (Table 1). To test the methylation frequency of the *FHIT* gene, we performed the MSP analysis of the *FHIT* gene in a panel of 207 tissue samples obtained from patients with CD. Strikingly, the *FHIT* gene was methylated in the vast majority (71%, 147 out of 207) of patients with CD samples (Figure 3b, right panel; Appendix A).

Next, we confirmed the methylation levels of the *FHIT* gene in selected patients with CD samples and we used quantitative MSP analysis (Figure 3c). The methylation level of *FHIT* was significantly increased in the tissues of patients with CD compared to that in normal colon tissues. Moreover, we also confirmed the CpG island methylation level of *FHIT* at the DNA sequences using bisulfite sequencing analysis in representative patient with CD samples and normal colon tissues (Figure 3d).

Interestingly, the *FHIT* gene showed dense methylation in CD samples compared to that in normal colon tissues. Taken together, these results strongly suggested that the *FHIT* candidate gene showed increasing DNA methylation levels in patient with CD samples compared with normal controls. To correlate promoter hypermethylation and FHIT expression in patients with CD, we examined FHIT protein expression by immunohistochemical staining. Immunopositivity for the FHIT was significantly decreased in colonic epithelial cells from patients with CD, whereas those of the normal colon exhibited strong staining in the intracytoplasmic and nuclear regions (Figure 4).

### 3.4. Functional Implication of the Hypermethylated Genes in CD as Analyzed by a Gene Network

It has been suggested that aberrant signaling pathways play a vital role in inflammatory development, resulting in dysregulation of the inflammatory responses in patients with IBD [27]. To understand the functional implications of hypermethylated genes in CD, we investigated the functional network of these hypermethylated genes in Figure 5, which is associated with biological and cellular pathways. Using significantly hypermethylated genes (top 200 genes) among a large set of differentially methylated genes in Figure 1 (Appendix A), we performed gene ontology (GO) analysis of our identified hypermethylated genes in CD using KEGG databases, which include a variety of biological processes, molecular functions, and cellular components.

Several pathways associated with the hypermethylated genes were related to the bleb assembly, regulation of actin filament-based processes, steroid metabolic processes, regulation of ion transport, protein localization to the cell periphery, positive regulation of insulin secretion, DAG and IP3 signaling, connective tissue development, blood vessel morphogenesis, peptidyl-serine dephosphorylation, aldosterone synthesis and secretion, cell junction organization, and positive regulation of binding (Figure 5a).

In addition, we also employed the GeneMANIA algorithm to predict network-based functional connections between genes that were differentially methylated in patients with CD samples. Hypermethylated genes in the CD samples (>2.5-fold increase in methylation; 19 genes in Figure 2a) were observed to interact with each other and were involved in several signaling pathways (Figure 5b). Our identified panel of hypermethylated candidate genes in CD could inactivate biological pathways associated with cell structure or cytoskeleton formation from the plasma membrane, such as pathways related to the bleb assembly and regulation of actin filament-based processes.

In addition, we also analyzed the gene network using hypomethylated genes in patients with CD. Interesting functional connections were found among hypomethylated genes in CD, including processes related to leukocyte activation involved in the immune response, lymphocyte proliferation, cytokine production, and signaling by interleukins (Appendix A), suggesting that there might be important biological roles for hypomethylated genes in the pathogenesis of CD. Overall, our data provide important information that has the potential to be used for developing therapeutic targets for patients with CD.

## 4. Discussion

Various signaling pathways involved in cancer development can be affected by this epigenetic event. The biological and clinical relevance of gene silencing regulated by hypermethylation has been comprehensively investigated in other human diseases, such as inflammatory diseases [8,28]. Recent studies have implicated gene-specific changes in DNA methylation in IBD pathogenesis [14,29,30,31]. In addition, we have previously reported that several gene loci associated with genes are cancer, specifically hypermethylated in colorectal cancer, and promoter hypermethylation could also be detected in patients with UC and CD [21,32]. Although these observations suggest that abnormal DNA methylation alterations might be useful as diagnostic or prognostic biomarkers for UC patients, there are few reports of epigenetic factors that have been altered by hypermethylation in IBD.

Genetic epidemiological data suggest that UC and CD share some, but not all, susceptibility loci [33]. However, a number of reports have made substantial progress in determining that the genetic variation in IBD subtypes is mostly overlapped [34]. Similar to genetic findings in IBD, we assume that a number of epigenetic disease associations may be shared by both UC and CD. Many reports have suggested that there are no major differences in the transcriptome profiles between UC and CD using meta-analyses of whole genome transcriptional profiling [35]. In addition, a study analyzing the DNA methylation profile of colonic mucosal biopsies of pediatric patients with IBD reported shared differentially methylated regions (DMRs) between CD and UC [36]. These reports agreed with our previous reports on independent methylation analyses of CD and UC. We did not comprehensively compare CD and UC, but a specific locus, *TCERG1L*, was frequently hypermethylated in both CD and UC [32,37]. Although IBD subtypes share genetic and epigenetic associations at the genome level, differential epigenetic changes or genetic changes provide multiple cellular pathways, such as T cell regulation for CD and cellular response to bacterial origin for UC, which may be involved in IBD pathogenesis [38].

This study has identified several convincing site-specific DNA methylation changes in genes that have not been widely described in CD but are associated with cellular pathways that are important to the disease. Here, we attempted to validate 19 genes that were hypermethylated in the tissues of patients with CD compared to normal colon tissues using the HumanMethylation 450K Beadchip platform. Two of the genes (*IL5RA* and *CCDC428*) were eliminated because no CpG islands were located in their promoter regions. To further filter down genes with increased methylation levels in patients with CD tissues compared to normal controls, the following experimental validation criteria were used based on the methods in our previous study [21]: (1) the gene was expressed in normal colon tissue, which we confirmed by IHC analysis for only the *FHIT* protein; (2) the gene had low or no methylation in normal colon tissues; and (3) the gene had methylation in samples from patients with CD. Here, we identified the best candidate gene, *FHIT*, which fulfilled the above criteria.

*FHIT* belongs to the histidine triad gene family, which encodes a hydrolase of Ap3A [39], and the *FHIT*-Ap3A enzyme-substrate complex plays a role as a tumor suppressor [40]. Since *FHIT* is located on chromosome 3, surrounding the common fragile site FRA3B, translocations and abnormal transcripts of *FHIT* frequently occur due to carcinogen-induced damage [41]. Aberrant transcripts of *FHIT* have been found in other kinds of tumors, such as gastric cancer [42], esophageal cancer [43], lung cancer [44], and colon carcinomas [45], indicating its potential role in suppressing carcinogenesis. *FHIT*^−/−^ mice were more prone than wild-type mice to developing carcinogen-induced tumors [46,47]. Moreover, recent studies have found that *FHIT* can also function as a tumor suppressor by inhibiting the epithelial-mesenchymal transition (EMT) and that loss or aberrant transcripts of *FHIT* may be associated with carcinogenesis [48,49]. Recently, detection of FHIT methylation identified by meta-analyses was shown to be useful for the early diagnosis of breast carcinoma [50] and NSCLC [51]. *FHIT* methylation has been identified in various types of cancer, including colon cancer; to the best of our knowledge, this is the first report implicating aberrant DNA hypermethylation of *FHIT* in IBD pathogenesis.

Further investigations of the genes we identified are required to define biological functions, as well as clinical relevance of hypermethylated genes in CD. We previously studied the promoter DNA methylation pattern of the *TCERG1L* gene, which is very frequently methylated in Colon cancer patients, in the blood samples of patients with CD to detect any disease-specific methylation. Indeed, we were able, using conventional MSP analysis, to detect promoter hypermethylation of *TCERG1L* in over half of the blood samples from patients with CD that we tested. We also confirmed the DNA methylation status of *TCERG1L* at the sequence level in patients with CD tissues and blood samples by bisulfite sequencing analysis [32]. A previous study led us to screen unknown genes whose transcriptional expression was regulated by hypermethylation in CD. This study has some limitations. The methylation analysis of *FHIT* was carried out in a large sample size of patients with CD, and we analyzed the clinical relevance of *FHIT* methylation in relation to CD clinical parameters (disease duration and location at diagnosis). Interestingly, patients with CD and *FHIT* methylation showed poor outcomes among patients with CD who had been diagnosed over a 5-year period (*n* = 88), but the difference was not statistically significant (Appendix A). However, our study provides clinically relevant evidence that the methylation of *FHIT* might be correlated with CD clinical parameters. Importantly, patients with CD with *FHIT* methylation diagnosed over a 5-year period exhibited poor outcomes suggestive of severe disease behavior and an increased risk of disease. Another limitation is the lack of transcriptome profiles of patients with CD. A correlation between methylation and transcriptional repression could be more effective in defining biomarkers for the diagnosis or prognosis of patients with CD than methylation changes alone. By elucidating DNA methylation alterations in CD, our study paves the way to a better understanding of the role of epigenetics in the pathogenesis of CD and provides direction for future research of additional *FHIT*-related diagnostic/prognostic or therapeutic approaches for CD.

## 5. Conclusions

In summary, this study provides important evidence that there is a set of DNA methylation alterations that are specific to patients with CD. More importantly, this study identified a promising methylation biomarker, the Fragile Histidine Triad (*FHIT*) gene, which could be useful for clinical surveillance for patients with CD. We provided evidence that the *FHIT* gene is hypermethylated in a disease-specific manner in a large sample of patients with CD and its protein level was downregulated in patients with CD, suggesting that its gene silencing by promoter methylation is correlated with its expression. Using our genome-wide DNA methylation profile of patients with CD, we also implicated that pathway analysis of the hypermethylated candidates further suggests putative molecular interactions relevant to IBD pathology. Future studies will focus on elucidating the biological role of FHIT in IBD pathogenesis, as this could be a therapeutic target.

## 6. Data Availability

Raw data are available on the Gene Expression Omnibus (GEO) website under the accession number GSE105798.

## Figures and Tables

**Figure 1 jcm-09-01338-f001:**
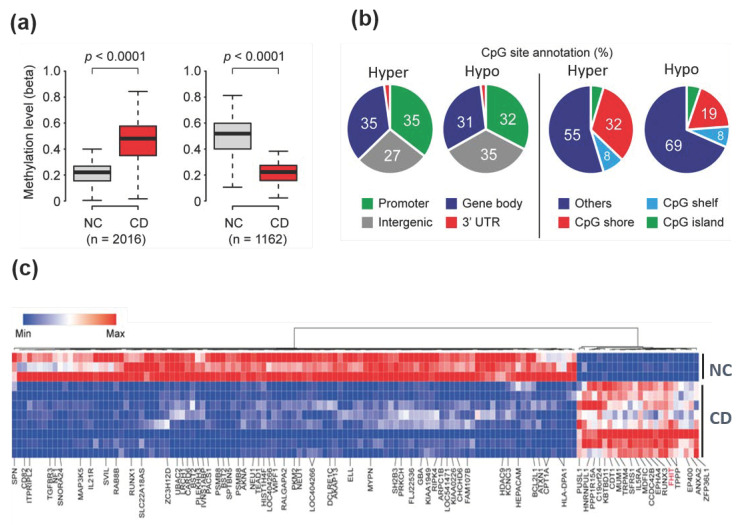
Genome-wide differentially methylated regions in colonoscopy samples from Crohn’s disease patients. (**a**) Genome-wide DNA methylation levels between patients with Crohn’s disease (CD) and normal control tissues were compared using the HumanMethylation450K BeadChip kit. A total of 2016 CpG sites were hypermethylated, while 1162 CpG sites were hypomethylated. *p*-values were calculated using the Mann–Whitney *U* test. (**b**) Venn diagram showing the proportion of genome-wide coverage of differentially methylated regions. Promoters, gene bodies, intergenic regions, and gene 3′ ends are shown (left panel). The number of methylated DNA loci in islands versus CpG shores (2-kb regions flanking CpG islands) and shelves (2-kb regions flanking shores) (right panel). (**c**) Heat map and hierarchical clustering dendrogram of differential gene methylation profiles between normal colon (NC) tissues and Crohn’s disease (CD) tissues.

**Figure 2 jcm-09-01338-f002:**
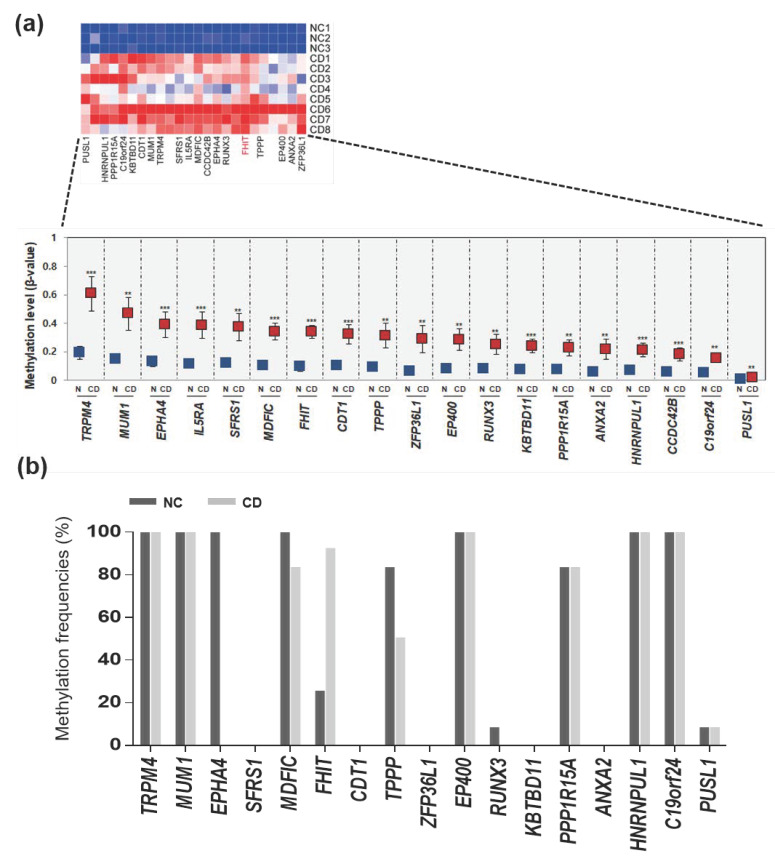
Hypermethylation pattern of candidate genes in patients with CD. (**a**) Heat map and hierarchical clustering dendrogram of 19 hypermethylated genes that exhibited > 2.5-fold increases in β-values in CD tissues compared to normal tissues. (**b**) Methylation frequencies of candidate genes in CD tissues (*n* = 12) compared to normal tissues (*n* = 12) determined by methylation-specific PCR (MSP).

**Figure 3 jcm-09-01338-f003:**
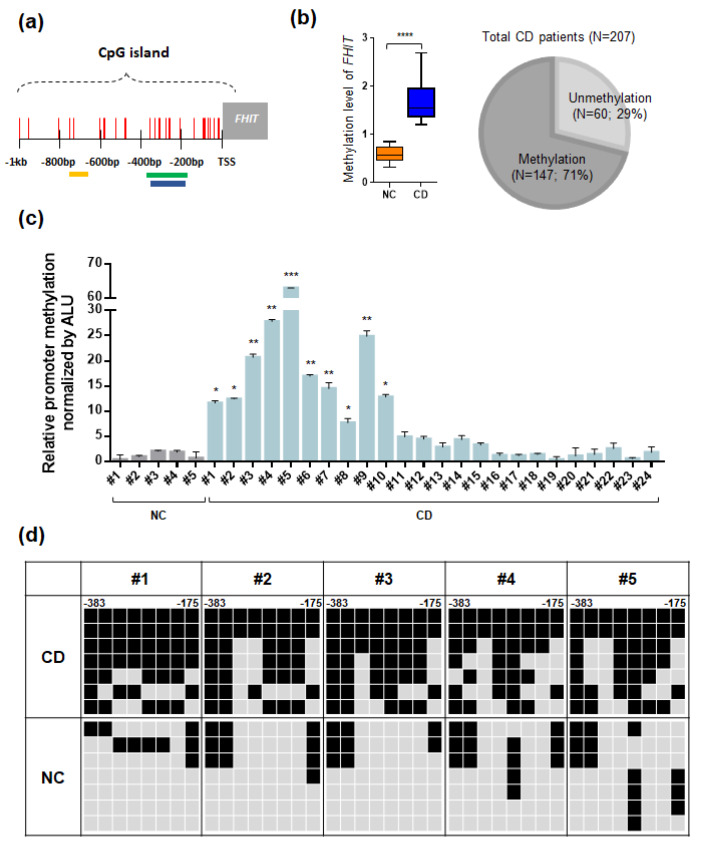
Summary of methylation analyses of the fragile histidine triad (*FHIT*) in CD tissues. (**a**) A schematic representation of the CpG islands in the *FHIT* gene promoter region. Each CpG site is indicated as a vertical red line, and the yellow, green, and blue boxes indicate the CpG probe site, amplicons for MSP, and bisulfite sequencing, respectively. (**b**) Methylation analysis of *FHIT* in patients with CD samples. Methylation level of *FHIT* between CD tissues and normal colon tissues (*n* = 12 each) and a Venn diagram indicates the methylation frequency of *FHIT* in a larger cohort of patients with CD (*n* = 207). (**c**) Quantitative MSP of the *FHIT* gene in selective ulcerative colitis (UC) patient samples and controls. All quantitative methylation levels were normalized by the *Alu* element. The statistical significance (*p* < 0.001) for the *FHIT* gene is shown between patients with CD samples. (**d**) Bisulfite sequencing analyses of the CpG islands in *FHIT* gene promoter regions. Bisulfite sequencing analyses were performed with representative patients with CD samples (CD; *n* = 5) and normal colon tissues (NC; *n* = 5). The location of CpG sites in the *FHIT* (upstream region from −383 to −175) relative to the transcription start sites (TSSs) of exon 1. Each box represents a CpG dinucleotide. Black boxes represent methylated cytosines and gray boxes represent unmethylated cytosines.

**Figure 4 jcm-09-01338-f004:**
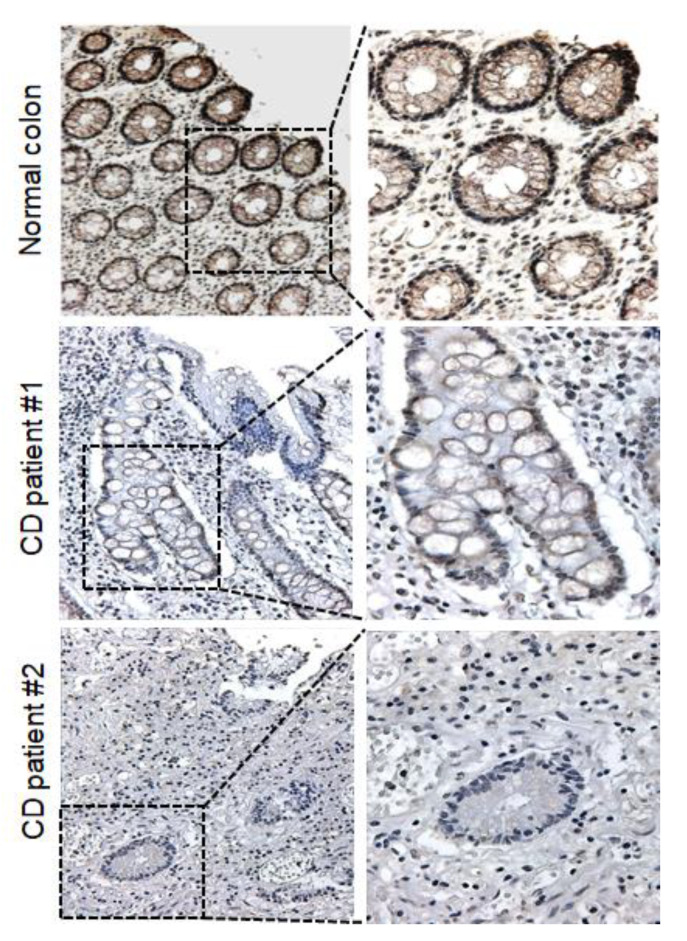
Protein expression levels of FHIT in CD tissue and normal colon tissue. Representative immunohistochemical analysis results showing FHIT expression in CD tissues and normal colon tissues. IHC analysis of FHIT was performed in both normal colon tissues (*n* = 5) and patients with CD tissues (*n* = 5). (Right images: 200×; left images: 400×).

**Figure 5 jcm-09-01338-f005:**
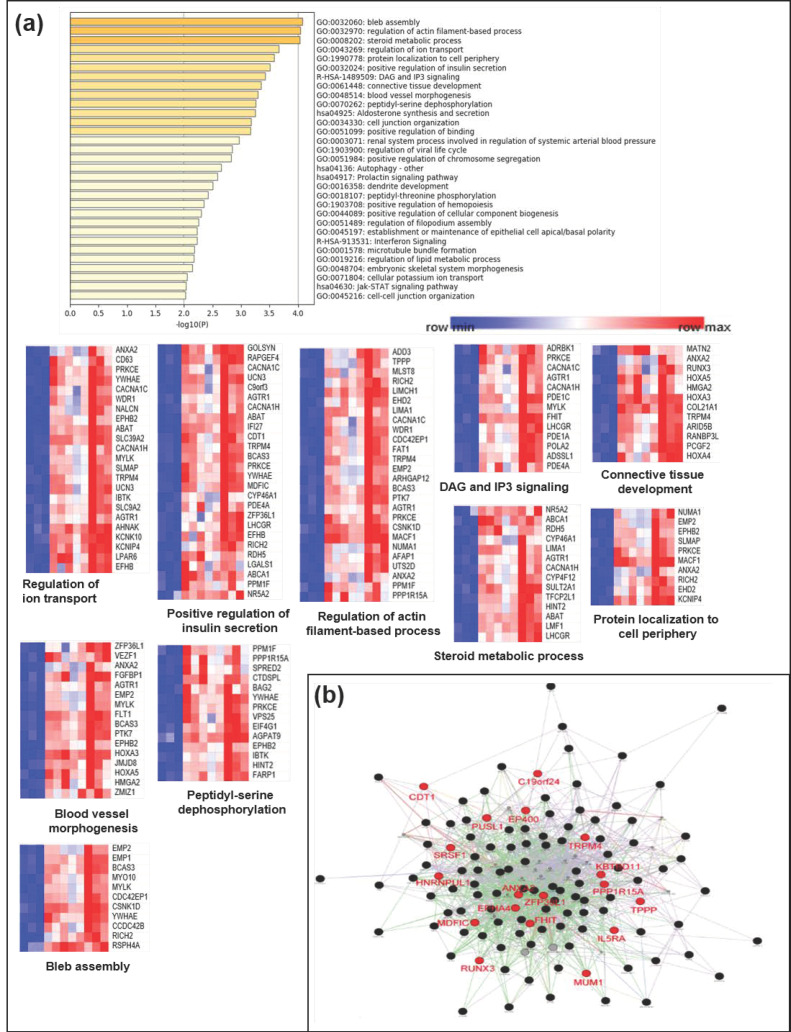
Biological functional implication of hypermethylated genes in CD. (**a**) Gene ontology (GO) analysis was conducted using Metascape and (**b**) GeneMANIA with genes that were proximal to the 143 hypermethylated genes (>2.5-fold increase). Each heatmap of the implicated biological pathways indicates those pathways significantly associated with the genes in the network. Similar terms tend to be clustered in the plot. The size of the circle depicts whether a given term is a more general GO term (larger) or a more specific term (smaller). The 19 hypermethylated candidate genes (>3-fold increase) are indicated in red.

**Table 1 jcm-09-01338-t001:** Basic characteristics of patients with CD in this study.

Characteristics	*n* = 207
Age (years)	
Median	33
Gender	
Male	152 (73.6%)
Female	55 (26.4%)
Duration from diagnosis to sampling	
≤1 year	26 (12.5%)
1–5 year	76 (37.0%)
≥5 year	105 (50.5%)
Months (median)	63
Disease location at diagnosis	
Small bowel alone	58 (27.9%)
Colon alone	23 (11.1%)
Small bowel and colon	126 (61.1%)
Disease behavior at diagnosis	
Inflammatory (B1)	130 (62.5%)
Stricturing (B2)	32 (15.9%)
Penetrating (B3)	45 (21.6%)

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
