# Peer review of "Genome-Wide Analysis of the DNA Methylation Profile Identifies the Fragile Histidine Triad (FHIT) Gene as a New Promising Biomarker of Crohn’s Disease"

_jcm, 2020, doi:10.3390/jcm9051338_

Round 1

Reviewer 1 Report

In this study, the authors analyzed DNA methylome to clarify the involvement of epigenetic alterations in the pathogenesis of Crohn’s disease (CD). They used Infinium BeadChip platform to analyze genome-wide DNA methylation in a series of specimens from CD patients and control individuals. They identified a series of genes aberrantly methylated in CD, and found frequent FHIT hypermethylation. They also found that genes aberrantly methylated in CD are associated with various signaling or biological function. The data shown in this manuscript are novel and of potential importance. I suggest several points which need to be addressed before acceptance.

The authors performed methylation-specific PCR (MSP) to validate the results of the BeadChip. Representative results of the MSP analysis should be shown.

Page 4, line 144: “UC” should be corrected to “CD”.

Page 5, line 204: The authors selected hypermethylated genes, ZFP36L1, ANXA2, EP400, FHIT, TPPP, IL5RA, MDFIC, MUM1, PUSL1, RUNX3, C19orf24, TRPM4, PPP1R15A, CDT1, SFRS1, EPHA4, CCDC42B, and HNRNPUL1 (18 genes), but they describe that they selected 19 candidate genes (lines 207, 208 and 210). Please check.

Among the 19 candidate hypermethylated genes, only FHIT was found to be prevalently hypermethylated in CD tissues. The authors should discuss the reason of the discrepancy between results of BeadChip and those of MSP.

To validate the inverse association between methylation and expression of FHIT, the authors performed immunohistochemical staining of FHIT as shown in Figure 4. The authors should describe how many samples were tested by immunohistochemistry. It is better to confirm negative or positive FHIT expression in multiple samples with or without FHIT gene methylation, if possible.

Page 9, line 272: In the sentence “we investigated the functional of network of these hypermethylated genes in Figure 5”, it is unclear how many genes were analyzed for the functional network analysis. Similarly, the list of hypermethylation genes analyzed in GO analysis is also unclear (line 273). If they represent the 2016 CpG hypermethylated probes (page 4, line 155 and Figure 1), the authors should provide the list of the genes as a supplementary information.

Reviewer 2 Report

In the manuscript by Kim, TO et al, the authors identify FHIT as a gene with promoter DNA hypermethylation in Crohn’s disease (CD). While DNA methylation alterations have been studied in ulcerative colitis, this study focuses on CD, which has been under-studied. The study uses an initial cohort of patients and genome-wide DNA methylation analysis to identify candidate genes with altered methylation in CD versus normal epithelium. The top hits are further verified in a larger group, with FHIT DNA hypermethylation being cofirmed in a second cohort of 203 CD samples. FHIT DNA hypermethylation is verified by both qMSP and bisulfite sequencing. Furthermore, by IHC the authors demonstrate that normal epithelium expresses FHIT protein, but this expression is reduced in epithelial cells in CD samples. Overall the manuscript is well-written and appropriate validation of the FHIT DNA methylation is performed. A few relatively minor concerns remain:

1) Of the 19 genes examined by MSP, are the differentially methylated probes from the 450K data in the same regions as the qMSP primers? It is clear that the qMSP primers were designed to assay DNA methylation in the CpG islands of these genes, but it is not clear where the differentially methylated probes were located. If the probes were not located in the promoter CpG islands, it may explain why so few validated.

2) Can the FHIT 450K probe with differential methylation be indicated in Figure 3A?

3) It is unclear how the data in the graph in Figure 3B is numerical but the data for the methylation status of the 203 samples is binary. Please include how the binary call was made. If it was done by gel-based MSP, a representative gel of a few samples in a supplemental figure would be sufficient.

4) The inclusion of the IHC data for FHIT is important as it demonstrates that DNA methylation of FHIT is associated with decreased protein expression in epithelial cells. In the figure legend it states that the images are representative. Please include the numbers of normal and CD samples in total that were assayed.

5) It is unclear which set of genes the analysis in section 3.4 (Figure 5) was performed on, the large set of differentially methylated genes or the smaller set with more stringent cut-offs? Please include this information.

Text edits:

Abstract
line 21: comma after epigenetic should be moved to after factors. “particularly epigenetic factors, can also significantly contribute”

Line 27: add ‘of a’ “bisulfite sequencing of a validation set”

B to indicate panel b in figure 3 is missing.
